# Associations between Conventional and Emerging Indicators of Dietary Carbohydrate Quality and New-Onset Type 2 Diabetes Mellitus in Chinese Adults

**DOI:** 10.3390/nu15030647

**Published:** 2023-01-27

**Authors:** Zhixin Cui, Man Wu, Ke Liu, Yin Wang, Tong Kang, Shuangli Meng, Huicui Meng

**Affiliations:** 1School of Public Health (Shenzhen), Shenzhen Campus of Sun Yat-sen University, Sun Yat-sen University, Shenzhen 518107, China; 2Shenzhen Health Development Research and Data Management Center, Shenzhen 518028, China; 3Guangdong Provincial Key Laboratory of Food, Nutrition and Health, Guangzhou 510080, China; 4Guangdong Province Engineering Laboratory for Nutrition Translation, Guangzhou 510080, China

**Keywords:** carbohydrate quality, dietary glycemic index, carbohydrate to fiber ratio, carbohydrate quality index, type 2 diabetes mellitus, Chinese population

## Abstract

Dietary glycemic index (GI), carbohydrate to fiber ratio (CF) and carbohydrate quality index (CQI) are conventional and emerging indicators for carbohydrate quality. We aimed to investigate the associations between these indicators and new-onset type 2 diabetes mellitus (T2DM) risk among Chinese adults. This prospective cohort study included 14,590 adults from the China Health and Nutrition Survey without cardiometabolic diseases at baseline. The associations between dietary GI, CF and CQI and T2DM risk were assessed using Cox proportional hazard regression analysis and dose–response relationships were explored using restricted cubic spline and threshold analysis. After a mean follow-up duration of 10 years, a total of 1053 new-onset T2DM cases occurred. There were U-shaped associations between dietary GI and CF and T2DM risk (both *P-nonlinear* < 0.0001), and T2DM risk was lowest when dietary GI was 72.85 (71.40, 74.05) and CF was 20.55 (17.92, 21.91), respectively (both *P-log likelihood ratio* < 0.0001). Inverse associations between CQI and T2DM risk specifically existed in participants < 60 y or attended middle school or above (both *P-trend* < 0.05). These findings indicated that moderate dietary GI and CF range and a higher dietary CQI score may be suggested for T2DM prevention in Chinese adults.

## 1. Introduction

Diabetes mellitus (DM) has become a major public health challenge in China [1]. The prevalence of DM in adults has increased from 10.9% in 2013 to 12.4% in 2018 [1]. China has been estimated as ranking first in the number of adults with DM in both 2021 and 2045 in the world [2]. Type 2 diabetes mellitus (T2DM) accounts for 90% of total DM [3].

Carbohydrate quality has always been considered an essential component of dietary patterns aimed to reduce T2DM risk [4]. Due to the numerous categories and complexity of carbohydrate-containing foods and diets, several indicators of carbohydrate quality, including dietary glycemic index (GI), carbohydrate to fiber ratio (CF) and carbohydrate quality index (CQI), have been proposed to rank carbohydrate-containing foods and diets in order to relate their quality to chronic disease risk and guide consumers in their food choices. As a conventional indicator of carbohydrate quality, GI was introduced since 1981 to rank carbohydrate-containing foods in terms of their postprandial glycemic response [5]. The concept of dietary GI was then created as an indicator of the carbohydrate quality of diet [6]. Although there has been plenty of evidence for the application of dietary GI on glycemic control in diabetic patients [7], associations between dietary GI and new-onset T2DM risk have been investigated in a considerable body of work with heterogeneous findings. Some meta-analyses and prospective studies have reported a positive association between dietary GI and T2DM risk [8,9,10], whereas others have reported no significant association [11,12,13,14,15]. The equivocal results may be attributed to issues related to methodology of dietary GI calculations or factors associated with inter-individual variability, such as sex, age, race and ethnic groups, background dietary patterns and food sources of dietary carbohydrates [11,13,16,17,18,19,20]. The majority of previous studies have focused on Western populations [11,12,13,15,19,21,22], and available data on the association between dietary GI and T2DM risk in the Chinese population are limited to only one study based on middle-aged Chinese women in Shanghai [23].

CF is an emerging indicator of carbohydrate quality, and evidence has shown that CF < 10: 1 may identify fiber-rich whole grain foods [24]. Research on the association between dietary CF and T2DM risk is scarce. The results from the Nurses’ Health Study have demonstrated that dietary CF is marginally associated with a higher risk of T2DM among US women [25]. Another cross-sectional study has also reported an inverse association between dietary CF and circulating adiponectin concentration in US women, indicating that dietary CF may relate to intermediate biomarkers of T2DM [26]. However, the association between dietary CF and T2DM risk in individuals other than US women remains unknown.

CQI is an emerging multi-component indicator of carbohydrate quality that incorporates dietary GI, fiber intake, ratio of whole grain to total grain and ratio of solid carbohydrate to total carbohydrate into a single comprehensive index [27]. A randomized trial has reported robust significant decreases in fasting concentrations of blood glucose and glycated hemoglobin across consecutive quintiles of increased CQI [28], indicating an inverse association between CQI and risk factors for T2DM development. However, prior to our study, only one cross-sectional study has investigated the association between CQI and T2DM risk and has reported no significant association [29]. Whether CQI is related to T2DM risk longitudinally has not yet been investigated.

Along with the increasing T2DM risk in the past decade, both the quantity and quality of dietary carbohydrate intakes have decreased in Chinese populations. In addition, food sources of carbohydrates have become more diverse, including not only carbohydrate-rich staple foods, but also sugar-sweetened beverages and pre-packaged foods enriched in sugar or refined carbohydrates [30,31,32]. It is still controversial whether indicators of dietary carbohydrate quality should be incorporated into dietary guidelines or nutrition labeling systems to guide food choices among Chinese populations in order to reduce T2DM risk [33]. Investigations on the associations between conventional and emerging indicators of carbohydrate quality and T2DM risk are required to provide evidence for the controversy.

The purpose of the current study was to investigate the associations between conventional and emerging indicators of dietary carbohydrate quality, including dietary GI, CF and CQI, and T2DM risk in a nationwide prospective cohort of Chinese adults. Our hypothesis was that higher dietary GI and CF values would be positively associated with T2DM risk, whereas a higher CQI score would be inversely associated with T2DM risk.

## 2. Materials and Methods

### 2.1. Study Population

Data of the current study were from the China Health and Nutrition Survey [34], which is an ongoing longitudinal household-based cohort study initiated in 1989 and 10 rounds have been completed between 1989 and 2015 [35,36]. Using an approach of multistage random cluster sampling, community-dwelling participants were selected from 9 provinces (including Liaoning, Jiangsu, Shandong, Hubei, Henan, Hunan, Guizhou, Guangxi, Heilongjiang) in each survey round between 1989 and 2011, and participants from 3 mega-cities (including Beijing, Shanghai and Chongqing) and 3 additional provinces (including Yunnan, Shaanxi and Zhejiang) were further selected under the same sampling strategy since 2011 and 2015, respectively [36,37]. Detailed information of the study design and methods of CHNS have been reported previously [35,36]. The CHNS study was conducted in accordance with the guidelines laid down in the Declaration of Helsinki and written informed consents were obtained from all participants. All study procedures of the CHNS study were approved by the Ethics Committee of the University of North Carolina at Chapel Hill (Project identification code 07-1963) and the National Institute for Nutrition and Health at the Chinese Center for Disease Control and Prevention (Project identification code 201524) [38,39].

This prospective cohort study was conducted by using data extracted from wave 1997 to 2015 of CHNS and a total of 33,314 participants were selected. In the current investigation, we excluded participants who were aged below 18 years old (*n* = 8992), missed all of the 3 consecutive 24 h dietary recall records (*n* = 3622), missed all the physical examination data (*n* = 98), were diagnosed with myocardial infarction, stroke, diabetes or cancer, or taking medicines to treat cardiometabolic disorders at baseline (*n* = 849), missed the follow-up survey (*n* = 3789), missed dietary records by food weighing methods (*n* = 129), had an unreasonable cumulative average of total energy intake (<800 or >4200 kcal/day for men, <500 or >3500 kcal/day for women, *n* = 283), or were during pregnancy or lactation at each survey round before the endpoint (*n* = 962) [40], a total of 14,590 adults were included in the final analysis (Appendix A).

### 2.2. Dietary Intake Data Collection and Assessment

Dietary intake data were collected from 3 consecutive 24 h dietary recall records (including two weekdays and one weekend) at the individual level, together with a food weighing method during the same period at the household level [41]. At the individual level, participants were asked to provide information on the type, amount, preparation method, time and location of each food consumed over the past 24 h under the guidance and supervision of trained interviewers [41]. The food weighing method at household level was used to confirm the amount of food intake and modify the records collected from the 24 h dietary recall at individual level [41]. The accuracy of assessing intakes of energy and nutrients by the 24 h dietary recall method and food weighing method has been validated in previous studies [42].

In the current study, the 3-day average intakes of dietary total energy and nutrients, including total carbohydrate, dietary fiber, protein, total fat, cholesterol, saturated fatty acid (SFA) and polyunsaturated fatty acid (PUFA) were calculated using information from the China Food Composition Tables [43,44,45,46] in every round from 1997 to 2011.

### 2.3. Calculation of Dietary GI, CF and CQI

Dietary GI values of each participant was calculated based on a previously established method of summing GI contribution of each carbohydrate containing food, which was calculated by multiplying GI value of the food item by the percentage of the available carbohydrate amount (g) of the specific food item relative to the total available carbohydrate amount (g) of the entire diet [47,48]. The GI value of each carbohydrate containing food item was obtained from either the China Food Composition Table [46] or International Tables of Glycemic Index and Glycemic Load Values 2008 [48].

Dietary CF value was calculated by dividing the dietary carbohydrate to fiber ratio. All nutrients and dietary GI and CF values were adjusted for total energy intake with the use of the residual method [49].

The CQI score was calculated on a basis of an established method [27]. In brief, participants were divided into quintiles according to each component of CQI, including dietary fiber intake (g/d), dietary GI, whole grain to total grain ratio and solid carbohydrate to total carbohydrate ratio. In terms of the dietary fiber intake, whole grain to total grain ratio and solid carbohydrate to total carbohydrate ratio, participants in the first through the fifth quintiles were assigned with a score ranging from 1 to 5 points, respectively. In contrast, the GI dimension was scored inversely, in which participants in the fifth through the first quintiles were assigned with a score ranging from 1 to 5 points, respectively. The CQI score was calculated by summing the points of the four components, and the range was from 4 to 20.

### 2.4. Ascertainment of T2DM

The primary outcome of the current study was new-onset T2DM risk occurring between 2000 and 2015 in the CHNS cohort. Health status information of participants was collected by trained interviewers through questionnaires including questions to confirm DM status of participants. Considering our participants for final analysis were not pregnant adults, the diagnosed DM cases were basically T2DM, being exclusive of gestational diabetes mellitus and type 1 diabetes mellitus which is of a low incidence in China and of early onset [50,51]. Cases of T2DM were defined as diagnosis of DM by physician, or currently using glucose control drugs or other corresponding treatments [52]. In addition, data collection in the 2009 wave comprised laboratory tests of fasting blood glucose and glycosylated hemoglobin (HbA1c), which were used as supplementary criteria for T2DM confirmation. In the 2009 wave, participants with either fasting blood glucose ≥ 7.0 mmol/L or HbA1c ≥ 6.5% were defined as T2DM cases [52,53]. The definition of T2DM was consistent with previous studies exploring the associations between dietary factors and new-onset T2DM risk with the use of the CHNS cohort [54,55]. If there were inconsistent records in the different survey waves of CHNS, the first wave record should prevail.

The baseline time was defined as the time when the participants first enrolled in the study during 1997–2015, which was the date of the dietary survey in the current analysis. In case of missing values at the date of dietary survey, the household interview date, which was very close to the date of dietary survey, was used as the baseline time. The follow-up time for each participant was calculated from baseline to the first T2DM diagnosis, the last wave before departure from the survey, death or the end of 2015 wave.

### 2.5. Assessment of Covariates

Participants were asked to provide information on sociodemographic, anthropometric and lifestyle characteristics, including age, sex, urbanization index, region, education level, smoking status, alcohol consumption and physical activity status, through validated questionnaires under instructions from trained interviewers [36]. The body weight and height of participants were measured in accordance with standard procedures by using calibrated equipment. Body mass index (BMI) was calculated as weight (kg) divided by the square of height (m^2^) [56]. Blood pressure was measured using a mercury sphygmomanometer for 3 consecutive times with 3–5 min intervals between two measurements [57]. The mean values of the three measurements of systolic blood pressure (SBP) and diastolic blood pressure (DBP) were used in analysis. Urbanization index ranging from 1 to 120 was calculated by using a scale consisting of 12 components, including population density, economic activity, traditional markets, modern markets, transportation infrastructure, sanitation, communications, housing, education, diversity, health infrastructure and social services [36], and was categorized in tertiles as low (33.5–56.3), moderate (56.3–80.6) or high (80.6–99.3). In the current study, region was geographically divided into northern or southern China with the Qinling Mountains and the Huaihe River as the demarcation line. Northern regions included Liaoning, Heilongjiang, Shandong and Henan provinces and Beijing and southern region included Jiangsu, Hubei, Hunan, Guizhou and Guangxi provinces and Shanghai as well as Chongqing [58]. Education levels were categorized into primary (primary school or lower), middle (middle school) or high (high school or above). Data on physical activity status were collected via a self-reported questionnaire assessing the time and intensity of occupational, transportational, household and leisure-time activities for each participant, and metabolic equivalent task hours per week (METs-h/week) were calculated accordingly for analysis [59]. Smoking and alcohol consumption was defined as having smoked or drunk alcohol since baseline to the end of follow-up.

### 2.6. Statistical Analysis

R software (version 3.6.3, The R Foundation for Statistical Computing, Vienna, Austria) was used for all statistical analyses. All statistical analyses were two-sided and statistical significance was accepted at *p* < 0.05. Cumulative average values of dietary intake, urbanization index, physical activity and BMI data were calculated and used in the analysis to reduce inter-individual variation and to capture long-term dietary patterns [60].

All participants were grouped into quintiles of energy-adjusted dietary GI or CF values or CQI scores, respectively. Cox proportional hazard regression models were used to assess the associations between dietary GI, CF and CQI values and T2DM risk, and to calculate the hazard ratios (HRs) and 95% confidence intervals (CIs) for the risk of T2DM. The first and fourth quintile of dietary GI or CF values were used as the reference group, respectively. The lowest quintile of dietary CQI values was used as the reference group. Model 1 adjusted for age (18–49, 50–54, 55–59, 60–64 or ≥65 y) and sex (male or female). Model 2 additionally adjusted for sociodemographic and lifestyle confounding factors, including education level (primary, middle or high), urbanization index (in tertiles as low, moderate or high), region (northern or southern), smoking status (yes or no), alcohol consumption (yes or no), BMI categories (<18.5 kg/m^2^ as low body weight, 18.5–23.9 kg/m^2^ as normal weight, 24.0–27.9 kg/m^2^ as overweight or ≥28.0 kg/m^2^ as obese) [61] and physical activity status (METs-h/week, in tertiles as low, moderate or high). Model 3 additionally adjusted for diet-related confounding factors including total energy (quintiles, kcal/d), cholesterol (quintiles, mg/d) and PUFA to SFA ratio (quintiles). Model 3 for dietary GI included total carbohydrate (quintiles, % energy) and fiber (quintiles, g/d) intakes as additional confounders. A test for linear trend was performed with the use of dietary GI, CF or CQI values as continuous variables by assigning the median values of quintiles to the variables in the Cox regression model.

Stratified analysis and potential effect modification were tested for the associations between dietary GI, CF and CQI values and T2DM risk by age (<60 y or ≥60 y), sex (female or male), BMI (<24.0 kg/m^2^ or ≥24.0 kg/m^2^), baseline hypertension (no (SBP < 140 mm Hg and DBP < 90 mm Hg) or yes (SBP ≥ 140 mm Hg or DBP ≥ 90 mm Hg)), urbanization index (median, <69.46 or ≥69.46), region (northern or southern), education level (“primary or lower” or “middle or above”), smoking status (no or yes), alcohol consumption (no or yes), physical activity status (median, <90.79 METs-h/week or ≥90.79 METs-h/week), total energy intake (median, <2105.44 kcal/d or ≥2105.44 kcal/d), total carbohydrate intake (median, <55.46% energy or ≥55.46% energy), cholesterol intake (median, <127.85 mg/d or ≥127.85 mg/d) and PUFA to SFA ratio (median, <1.15 or ≥1.15).

The potential for nonlinearity and dose–response relationship of the associations between dietary GI and CF values and the risk of T2DM was explored using restricted cubic spline (RCS) with 4 knots (5th, 35th, 65th, 95th), and dietary GI = 70.0 and dietary CF = 25.4 were set as the reference where the hazard ratio of T2DM was 1, respectively. Threshold analysis was used by applying two-piecewise Cox regression models adjusting the same confounders as model 3 to identify the inflection point of non-linear relationship between dietary GI and CF values and T2DM risk.

## 3. Results

### 3.1. Sociodemographic, Anthropometric and Lifestyle Characteristics of Study Participants at Baseline

During a mean follow-up of 10 years, a total of 14,590 participants were included in the current analysis and 1053 (7.3‰ person-year) participants developed new-onset T2DM. The sociodemographic, anthropometric and lifestyle characteristics of participants according to quintiles (Q) of dietary GI and CF are summarized in Table 1. The mean age of participants was 45 ± 15 years, 50.7% were males, and the mean BMI was 23.2 ± 3.3 kg/m^2^ (Table 1). The median dietary GI, CF and CQI of all participants were 69.9 (65.2, 73.8), 27.8 (21.5, 35.6) and 8.0 (7.0, 10.0), respectively (Table 1). The participants with higher dietary GI values were more likely to be younger adults, male participants, have higher BMI, live in northern China, smoke, drink alcohol, be physically active and have dietary intakes high in total carbohydrate and PUFA to SFA ratio, and less likely to have baseline hypertension, be educated, live in high urbanized regions or have higher intakes of fat, protein, fiber or cholesterol in comparison to participants with lower dietary GI values (all *P* < 0.05) (Table 1). The participants with higher CF values were more likely to be male participants, have lower BMI, live in southern China, smoke, be physically active and have high dietary intake of energy and total carbohydrate intakes, and less likely to have baseline hypertension, be educated, live in high urbanized regions, drink alcohol and have dietary intakes high in fat, protein, dietary fiber, cholesterol and PUFA to SFA ratio compared to participants with lower CF values (all *P* < 0.05) (Table 1).

### 3.2. Associations between Dietary GI, CF and CQI Values and T2DM Risk

In the Cox regression model adjusted for sociodemographic and lifestyle factors, there was an inverse association between dietary GI values and T2DM risk (model 2 for dietary GI; *P-trend* = 0.0023) (Table 2). In the fully adjusted model, higher dietary GI values were significantly associated with a reduced risk of T2DM, especially for Q4 (model 3 for dietary GI; Q4 compared with Q1, HR = 0.54; 95% CI: 0.43, 0.67; *P-trend* = 0.0024) (Table 2). When Q4 of dietary GI was set as the reference in the fully adjusted model, the adjusted HRs (95% CIs) of T2DM risk were 1.86 (1.49, 2.33) for Q1, 1.24 (1.01, 1.51) for Q2 and 1.34 (1.10, 1.64) for Q5, respectively, in comparison to Q4 (model 3 for dietary GI; *P-trend* = 0.0024) (Table 2). These results indicated a U-shaped relationship between dietary GI values and T2DM risk in our participants.

In the Cox regression model adjusted for age and sex, dietary CF value was inversely associated with T2DM risk when Q1 was set as reference, especially for Q4 (model 1 for dietary CF; Q4 compared with Q1, HR = 0.49; 95% CI: 0.35, 0.70; *P-trend* = 0.0071) (Table 2). Dietary CF values were not significantly associated with T2DM risk in model adjusted for sociodemographic and lifestyle factors and fully adjusted model (models 2 and 3 for dietary CF; both *P-trend* > 0.05) (Table 2).

No significant association between CQI and T2DM risk was observed in any model (models 1, 2 and 3 for CQI; all *P-trend* > 0.05) (Table 2).

### 3.3. Associations between Dietary GI, CF and CQI Values and T2DM Risk on the Basis of Potential Effect Modifiers

There was significant effect modification of the associations between dietary GI and T2DM risk by BMI (*P-interaction =* 0.0259), urbanization index (*P-interaction =* 0.0068) and dietary PUFA to SFA ratio (*P-interaction =* 0.0069) (Table 3). Dietary GI was inversely associated with the risk of T2DM in strata of BMI ≥ 24.0 kg/m^2^ (Q5 compared with Q1: HR = 0.52; 95% CI: 0.38, 0.71; *P-trend* = 0.0002), urbanization index < 69.46 (Q5 compared with Q1: HR = 0.45; 95% CI: 0.27, 0.76; *P-trend* = 0.0004) or dietary PUFA to SFA ratio ≥ 1.15 (Q5 compared with Q1: HR = 0.55; 95% CI: 0.40, 0.76; *P-trend* < 0.0001), respectively (Table 3). There was no evidence of effect modification of the associations between dietary GI and T2DM risk by other potential effect modifiers (all *P-interactions* > 0.05) (Table 3 and Appendix A). Results of the stratified analysis with the fourth quintile of dietary GI as the reference group were shown in Appendix A.

The results showed that BMI significantly modified the association between dietary CF and T2DM risk which was not a significant association in each strata (Table 3). There was no significant effect modification of the associations between dietary CF and T2DM risk by any other potential effect modifiers (all P-interactions > 0.05) (Table 3 and Appendix A). The results of the stratified analysis with the fourth quintile of dietary CF as the reference group were shown in Appendix A.

There was significant effect modification of the associations between CQI values and T2DM risk by age (*P-interaction =* 0.0324) and education level (*P-interaction =* 0.0069). CQI values were inversely associated with the risk of T2DM in strata of age < 60 y (Q5 compared with Q1: HR = 0.63; 95% CI: 0.50, 0.80; *P-trend* = 0.0015) and education level with middle school or above (Q5 compared with Q1: HR = 0.46; 95% CI: 0.34, 0.64; *P-trend* < 0.0001). The results also showed that dietary cholesterol intake significantly modified the association between dietary CQI values and T2DM risk which was not a significant association in each strata (Appendix A). There was no evidence of other potential effect modifiers (all *P-interactions >* 0.05) (Table 3 and Appendix A).

### 3.4. The Dose–Response Relationship between Dietary GI, CF and CQI Values and the Risk of T2DM

RCS analysis demonstrated U-shaped associations between dietary GI and T2DM risk (Figure 1A, *P-total* < 0.0001, *P-nonlinear* < 0.0001). Threshold analysis demonstrated that the risk of T2DM was lowest when dietary GI was 72.85 (71.40, 74.05) (Table 4, *P-log likelihood ratio* < 0.0001). Dietary GI was inversely associated with T2DM risk with the dietary GI range of 18.50–72.85 (HR = 0.95; 95% CI: 0.93, 0.96; *P* < 0.0001) (Table 4), and was positively associated with T2DM risk with the range of 72.85–88.22 (HR = 1.11, 95% CI: 1.07, 1.16; *P* < 0.0001) (Table 4).

A similar U-shaped pattern was observed for CF (Figure 1B, *P-total* < 0.0001, *P-nonlinear* < 0.0001). Threshold analysis demonstrated that the risk of T2DM was lowest when CF was 20.55 (17.92, 21.91) (Table 4, *P-log likelihood ratio* < 0.0001). CF was inversely associated with T2DM risk with the CF range of 3.07–20.55 (HR = 0.94; 95% CI: 0.91, 0.96; *P* < 0.0001) (Table 4), and was positively associated with T2DM risk with the range of 20.55–221.15 (HR = 1.03; 95% CI: 1.02, 1.04; *P* < 0.0001) (Table 4).

## 4. Discussion

To better define dietary carbohydrate quality to improve diet quality and human health as well as environmental sustainability, several indicators or metrics have been proposed to classify high-quality carbohydrate food or diet [62]. There is not currently any universal consensus on which indicator or metric should be incorporated into dietary guidelines and Nutrient Facts panel labeling intended to help reduce cardiometabolic disease risk [62]. Although dietary GI has been advocated by some research groups as a conventional indicator to represent carbohydrate quality, studies investigating the associations between dietary GI and T2DM risk have generated heterogeneous findings [11,12,13,14,15,19,20,21,22,23]. A limited prospective cohort study has assessed the associations between emerging indicators of carbohydrate quality, including dietary CF and CQI score and T2DM risk [25]. The current study was conducted to address these research gaps by assessing the associations between dietary GI, CF and CQI values and T2DM risk in a nationwide prospective cohort of Chinese adults. Data of the current study demonstrated U-shaped associations between dietary GI and CF values and T2DM risk in Chinese adults, and BMI, dietary PUFA: SFA ratio and urbanization index significantly modified the associations between dietary GI and T2DM risk. Inverse associations between CQI and T2DM risk were observed in participants < 60 y or who attended middle school or above.

In our study, a U-shaped association between dietary GI and the risk of T2DM was observed. Compared with the fourth quintile of dietary GI (71.4–74.8), the risk of T2DM was significantly higher in the fifth (74.8–88.2) quintile. The results of the threshold analysis further confirmed this finding. These results were in agreement with several observational studies and meta-analyses which have reported positive associations between dietary GI and T2DM risk [9,10,19,20,22,23]. The mechanism has been attributed, in part, to the higher postprandial glucose concentrations induced by high GI diets, resulting in greater insulin demand and therefore pancreatic beta cell exhaustion [63,64]. High GI diets have also been reported to directly promote insulin resistance via increasing late-postprandial free fatty-acid release [63,64,65].

In the U-shaped relationship, compared with the fourth quintile of dietary GI (71.4–74.8), the risk of T2DM was significantly higher in the first (18.5–63.9) quintile, contrary to our hypothesis. This finding was specifically observed in participants with BMI ≥ 24 kg/m^2^, higher dietary PUFA: SFA or who lived in areas with a lower urbanization index, and was also confirmed in the threshold analysis. These findings were somewhat unexpected because data from a prospective cohort study conducted in Shanghai have reported a 21% increased risk of T2DM in the largest quintile of dietary GI compared with the lowest [23]. In addition, low GI diets have been advocated to have independent beneficial effects for T2DM prevention by some research groups and consortium [9,66]. However, the OmniCarb study has reported that a high carbohydrate, low GI diet decreased insulin sensitivity and increased fasting glucose level compared with a high carbohydrate, high GI diet [67]. Another randomized intervention trial has also reported increased fasting glucose concentrations after consumption of a low-GI diet compared with a high-GI diet [68]. In addition, several studies have reported associations between GI values and gut microbial metabolites, such as γ-butyrobetaine, hippurate and 4-hydroxyhippuric acid [69,70], which have been linked to glucose metabolism, insulin resistance and risk for T2DM [71,72]. Therefore, the interplay between carbohydrate intake and gut microbiota may also partially contribute to the relationship between dietary GI values and T2DM risk. The reason for the inverse association observed in our study was not obvious, and potential mechanisms need to be further explored.

Despite the U-shaped relationship between dietary GI and T2DM risk observed in our study or positive associations reported in other studies, no significant association has also been reported previously in several studies [11,12,13,14]. These equivocal results may be partially attributable to inter-individual variances of participants, such as race [11,12,13,14,15,19,20,21,22,23], age [11,12,13,14,15,19,20,21,22,23] or differences in dietary intakes of other nutrients. Of note, dietary GI values of the majority of previous studies were in the first and second quintiles of dietary GI values in the current study [12,13,14,19], and only two studies have used residual method to avoid extraneous variance from total energy intake [20,23]. Therefore, we cannot rule out the possibility that these factors may contribute to the discordant results. In addition, dietary GI may not be precise enough for guiding food choices. Firstly, large intra- and inter-individual variability in food GI-value assessment has been reported in several studies [16,73]. Secondly, the meal or dietary GI can be influenced by other factors, such as nutrients consumed concurrently and the macronutrient composition of the prior meal [74,75]. Finally, a clinical trial has reported that the meal’s or dietary GI values can be overestimated using the current calculation formula [47]. These reasons may also account for the equivocal results.

Our current study found a non-linear and U-shaped association between dietary CF values and T2DM risk. When dietary CF was >20.55, dietary CF was positively associated with T2DM risk, which was consistent with results from the Nurses’ Health Study reporting a marginally positive association between CF and the risk of T2DM (RR = 1.09; 95% CI: 1.00, 1.20; *P* = 0.040) [25]. The potential underpinning mechanism may be, in part, attributed to the associations between higher dietary CF intake with lower adiponectin levels in circulation [26], which have been reported to relate to increased risk of T2DM [76]. When dietary CF was <20.55, there was an inverse association between dietary CF and T2DM risk, in which a significantly increased risk of T2DM was observed in the first quintile of CF compared with the fourth quintile. The potential underlying mechanism responsible for the inverse association has yet to be elucidated. In comparison to the fourth quintile, the lower dietary CF of the participants in the first quintile was due to both lower carbohydrate intake (48.8 ± 11.8% energy for the first quintile of CF vs. 56.8 ± 9.6% energy for the fourth quintile of CF, *P* < 0.0001) and higher dietary fiber intake (18.2 ± 7.6, g/d for the first quintile of CF vs. 9.0 ± 1.6, g/d for the fourth quintile of CF, *P* < 0.0001). A recent study based on CHNS cohort has reported a U-shaped association between dietary carbohydrate intake and T2DM risk in Chinese adults [77], indicating that the increased risk of T2DM in the lowest quintile of dietary CF in comparison to fourth quintile observed in our study may be partially explained by low carbohydrate intake.

Although there was no significant association between CQI and T2DM risk in all participants, we found that a higher CQI score, which reflected higher carbohydrate quality, was associated with a lower risk of T2DM in participants younger than 60 y or who had education level no less than middle school. The inverse association may be partially attributed to the ability to adhere to diets with a higher carbohydrate quality, as reflected by higher CQI scores, to reduce fasting blood concentrations of glucose and glycated hemoglobin [28]. Other than GI and CF, CQI is a multi-component indicator of carbohydrate quality incorporating dietary GI, fiber intake, ratio of whole grain to total grain and ratio of solid carbohydrate to total carbohydrate into a single index [27]. Therefore, it is a more comprehensive index than GI and CF. Prior to our study, there was only one cross-sectional study assessing the association between CQI and T2DM risk in 12,027 adults aged 19–64 years from the Korea National Health and Nutrition Examination Survey [29]. Inconsistent with our findings, this study reported no significant association between CQI and prevalence of T2DM. In the setting of a prospective cohort study, our study added new information to the literature supporting the long-term longitudinal association between higher CQI scores and reduced T2DM risk. CQI was introduced in the Seguimiento Universidad de Navarra (SUN) Project [27] for the first time in 2014. The majority of studies assessing the associations between CQI and cardiometabolic disease risk have been conducted using data from the SUN Project. Therefore, data from other cohorts are still required to confirm the ability of CQI to represent dietary carbohydrate quality and whether a high CQI score is related to better cardiometabolic health.

There are several strengths of the current study. The prospective design with long follow-up time and large-scale coverage of Chinese adults from 15 provinces or mega-cities enables the assessment of longitudinal association between indicators of dietary carbohydrate quality and T2DM risk in healthy Chinese adults who were free from cardiometabolic diseases at baseline. The quality of dietary intake data is guaranteed because three consecutive 24 h dietary recall records were used to reduce recall bias, and a food weighing method at household level was used to confirm the amount of dietary intake. The sociodemographic, anthropometric and lifestyle information of the participants were collected under strict quality control.

There are also some limitations in this study. The 24 h dietary recall records may not reflect long-term dietary intake as accurately as food-frequency questionnaires. However, we used the cumulative average dietary intake data as the exposure values to better represent the long-term diet. In our study, the T2DM cases were not confirmed by standard oral glucose tolerance test, which is impractical to conduct in such a large-scale cohort study. This issue was partially overcome with the use of biochemical test data in the 2009 wave to ascertain the T2DM cases. Although we have adjusted for potential confounders in our multivariable models, we cannot rule out the possibility of other potential residual confounders, which is a similar limitation as in other observational studies. Due to the observational nature of our study, mechanisms underpinning the U-shaped associations between dietary GI and CF values and T2DM risk were not explored. In addition, in terms of ascertainment of T2DM cases, when there were inconsistent records in the different waves of follow-up, the records of the first wave were used in the analysis to reflect incident risk of T2DM. Although a total of 596 participants had at least one record of being free of T2DM in later survey waves of CHNS after being diagnosed with T2DM, the first diagnosis of T2DM was used to reduce bias from changes of lifestyle or other interventions after the diagnosis that may undermine the association between dietary carbohydrate quality indicators and the risk of T2DM.

## 5. Conclusions

In conclusion, there were U-shaped associations between dietary GI and CF values and T2DM risk, indicating that both dietary GI and CF should be controlled in a moderate range for T2DM prevention in Chinese adults. BMI, dietary PUFA: SFA and urbanization index significantly modified the associations between dietary GI and T2DM risk. Inverse associations between CQI and T2DM risk specifically existed in participants < 60 y or who attended middle school or above. These findings, if replicated in a larger cohort, add new information to the current literature suggesting that there should be a reassessment of relationships between indicators of dietary carbohydrate quality and T2DM risk and careful consideration of whether these indicators should be incorporated into dietary guidelines and Nutrient Facts panel labeling intended to help reduce T2DM risk in the Chinese population.

## Figures and Tables

**Figure 1 nutrients-15-00647-f001:**
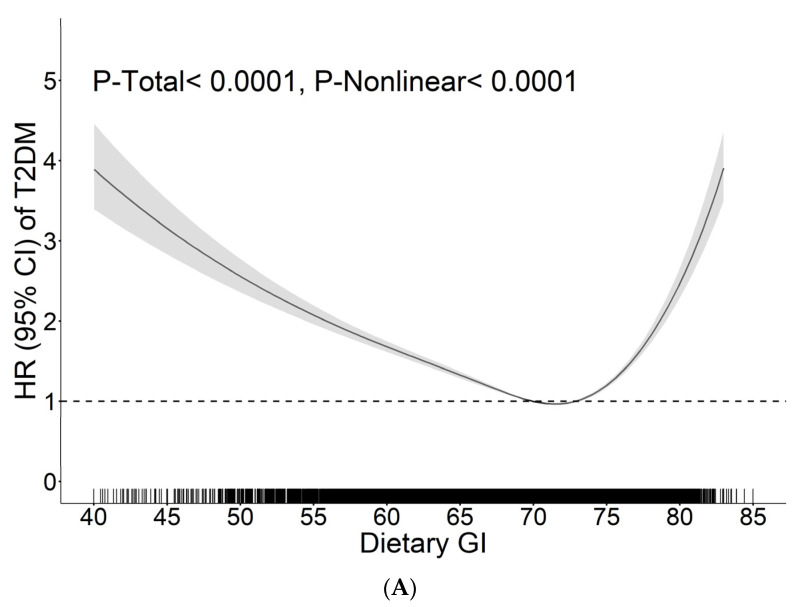
The dose–response relationship between dietary GI and CF values and T2DM risk in 14,590 Chinese adults of the China Health and Nutrition Survey 1997–2015 ^1^. ^1^ The potential for nonlinearity and dose–response relationship of the associations between dietary GI and CF values and the risk of T2DM was explored using restricted cubic spline with four knots (5th, 35th, 65th, 95th). Black solid lines were adjusted HRs of T2DM risks associated with dietary GI (**A**) or CF (**B**) values. Gray shade areas showed corresponding 95% CIs. Dotted lines indicated reference lines for no association (HRs of T2DM risk were 1) where dietary GI was 70.0 and dietary CF was 25.4. The rugs showed fraction of participants with different dietary GI ranging of 40–85 and CF ranging of 0–80, respectively. The HR of T2DM risk associated with dietary CF value was adjusted for age, sex, education level, urbanization index, region, smoking status, alcohol consumption, BMI, physical activity status, total energy, cholesterol and PUFA to SFA ratio and the HR of T2DM risk associated with dietary GI value was additionally adjusted for total carbohydrate and fiber intakes. Abbreviations: BMI, body mass index; CF, carbohydrate to fiber ratio; CI, confidence intervals; GI, glycemic index; HR, hazard ratio; PUFA, polyunsaturated fatty acid; SFA, saturated fatty acid; T2DM, type 2 diabetes mellitus.

**Table 1 nutrients-15-00647-t001:** Baseline sociodemographic, anthropometric and lifestyle characteristics of participants were presented according to quintiles of dietary GI and CF values in 14,590 Chinese adults of the China Health and Nutrition Survey 1997–2015 ^1^.

Variables	Total	Quintiles of Dietary GI Value			Quintiles of Dietary CF Value		
		Q1	Q3	Q5	Q1	Q3	Q5
*n*	14,590	2918	2918	2918	2918	2918	2918
Age, years	45 ± 15	46 ± 17	45 ± 14	45 ± 15	46 ± 14	44 ± 14	46 ± 17
Male, *n* (%)	7402 (50.7)	1384 (47.4)	1510 (51.7)	1554 (53.3)	1416 (48.5)	1468 (50.3)	1516 (52.0)
BMI, kg/m^2^	23.2 ± 3.3	23.0 ± 3.4	23.2 ± 3.2	23.4 ± 3.3	23.8 ± 3.5	23.4 ± 3.2	22.4 ± 3.2
Baseline hypertension, *n* (%)	3095 (21.2)	647 (22.2)	597 (20.5)	621 (21.3)	699 (24.0)	627 (21.5)	572 (19.6)
Education level, *n* (%)							
Primary	6860 (47.0)	1182 (40.5)	1417 (48.6)	1420 (48.7)	1004 (34.4)	1499 (51.4)	1568 (53.7)
Middle	4145 (28.4)	788 (27.0)	825 (28.3)	945 (32.4)	842 (28.9)	808 (27.7)	810 (27.8)
High	3585 (24.6)	948 (32.5)	676 (23.2)	553 (19.0)	1072 (36.7)	611 (20.9)	540 (18.5)
Urbanization index, *n* (%)							
Low	4846 (33.2)	629 (21.6)	915 (31.4)	1433 (49.1)	582 (19.9)	1171 (40.1)	1088 (37.3)
Moderate	4881 (33.5)	918 (31.5)	999 (34.2)	939 (32.2)	856 (29.3)	959 (32.9)	1007 (34.5)
High	4863 (33.3)	1371 (47.0)	1004 (34.4)	546 (18.7)	1480 (50.7)	788 (27.0)	823 (28.2)
Region							
Northern	6066 (41.6)	777 (26.6)	1125 (38.6)	1823 (62.5)	1361 (46.6)	1518 (52.0)	621 (21.3)
Southern	8524 (58.4)	2141 (73.4)	1793 (61.4)	1095 (37.5)	1557 (53.4)	1400 (48.0)	2297 (78.7)
Smoking status, *n* (%)							
No	9133 (62.6)	1973 (67.6)	1772 (60.7)	1804 (61.8)	1984 (68.0)	1778 (60.9)	1812 (62.1)
Yes	5457 (37.4)	945 (32.4)	1146 (39.3)	1114 (38.2)	934 (32.0)	1140 (39.1)	1106 (37.9)
Alcohol consumption, *n* (%)							
No	7478 (51.3)	1670 (57.2)	1367 (46.8)	1627 (55.8)	1509 (51.7)	1456 (49.9)	1620 (55.5)
Yes	7112 (48.7)	1248 (42.8)	1551 (53.2)	1291 (44.2)	1409 (48.3)	1462 (50.1)	1298 (44.5)
Physical activity status, METs-h/week	103.9 ± 84.0	89.5 ± 79.6	102.4 ± 83.0	118.4 ± 86.7	90.7 ± 76.1	114.3 ± 88.1	103.9 ± 81.9
Total energy intake, kcal/d ^2^	2106.2 ± 13.4	2105.8 ± 17.9	2106.8 ± 12.5	2105.8 ± 10.5	2104.8 ± 19.8	2106.3 ± 10.8	2107.4 ± 11.0
Total carbohydrate intake, % energy ^2^	55.1 ± 11.0	51.4 ± 12.2	54.4 ± 9.9	60.0 ± 10.2	48.8 ± 11.8	56.8 ± 10.4	58.8 ± 9.4
Total dietary fiber intake, g/d ^2^	11.6 ± 5.5	12.5 ± 8.2	11.1 ± 4.5	11.5 ± 4.2	18.2 ± 7.6	11.0 ± 2.1	6.7 ± 1.5
Fat intake, % energy ^2^	31.9 ± 10.3	34.6 ± 11.2	32.7 ± 9.5	27.6 ± 10.1	36.5 ± 11.3	30.4 ± 9.9	29.2 ± 8.9
Cholesterol intake, mg/d ^2^	153.3 ± 135.5	190.3 ± 152.5	158.6 ± 135.6	109.6 ± 125.7	171.6 ± 137.9	138.4 ± 133.4	155.7 ± 130.1
PUFA to SFA ratio ^2^	1.2 ± 0.7	1.2 ± 0.7	1.2 ± 0.6	1.4 ± 0.7	1.3 ± 0.6	1.4 ± 0.7	1.1 ± 0.6
Protein intake, % energy ^2^	12.2 ± 2.5	13.1 ± 3.3	12.0 ± 2.2	11.9 ± 2.0	13.5 ± 3.1	12.1 ± 2.0	11.4 ± 2.4
Dietary GI ^2^	69.9 (65.2, 73.8)	60.8 (57.9, 62.5)	69.9 (69.2, 70.7)	77.0 (75.8, 78.4)	66.6 (61.5, 71.0)	71.8 (67.9, 75.2)	69.0 (63.6, 73.3)
CF ^2^	27.8 (21.5, 35.6)	25.1 (16.5, 39.3)	27.9 (21.9, 35.3)	28.6 (24.4, 35.0)	16.3 (13.4, 18.3)	27.8 (26.6, 29.0)	45.1 (41.2, 51.8)
CQI	8.0 (7.0, 10.0)	10.0 (9.0, 12.0)	8.0 (7.0, 10.0)	7.0 (5.0, 8.0)	11.0 (9.0, 12.0)	9.0 (7.0, 10.0)	7.0 (6.0, 8.0)

^1^ Data are presented as mean ± SD, median (P_25_, P_75_) or *n* (%). Abbreviations: BMI, body mass index; CF, carbohydrate to fiber ratio; CQI, carbohydrate quality index; GI, glycemic index; MET, metabolic equivalent task hour; PUFA, polyunsaturated fatty acid; Q, quintiles; SD, standard deviation; SFA, saturated fatty acid. ^2^ Dietary nutrients, GI and CF were energy adjusted using the residual method.

**Table 2 nutrients-15-00647-t002:** Associations between dietary GI, CF and CQI values and T2DM risk in 14,590 Chinese adults of the China Health and Nutrition Survey 1997–2015 ^1^.

Variables	HR (95% CI) of Quintiles of Carbohydrate Quality Indicators					*P-Trend*
	Q1	Q2	Q3	Q4	Q5	
Dietary GI						
*n*	2918	2918	2918	2918	2918	
range	(18.5, 63.9)	(63.9, 68.3)	(68.3, 71.4)	(71.4, 74.8)	(74.8, 88.2)	
Median	60.8	66.3	69.9	73.0	77.0	
Cases (incidence rate, ‰ person-year)	184 (8.95)	224 (7.39)	216 (6.35)	212 (6.47)	217 (8.37)	
Model 1 ^2^	1.00 (Ref)	0.66 (0.54, 0.80)	0.56 (0.46, 0.68)	0.58 (0.47, 0.70)	0.85 (0.70, 1.04)	0.15
Model 2 ^2^	1.00 (Ref)	0.63 (0.52, 0.78)	0.54 (0.44, 0.66)	0.52 (0.42, 0.64)	0.73 (0.58, 0.91)	0.0023
Model 3 ^2^	1.00 (Ref)	0.66 (0.54, 0.81)	0.57 (0.46, 0.70)	0.54 (0.43, 0.67)	0.72 (0.57, 0.91)	0.0024
Model 1 ^3^	1.74 (1.42, 2.12)	1.14 (0.95, 1.38)	0.97 (0.80, 1.17)	1.00 (Ref)	1.48 (1.23, 1.79)	0.15
Model 2 ^3^	1.93 (1.56, 2.39)	1.22 (1.00, 1.49)	1.04 (0.86, 1.27)	1.00 (Ref)	1.40 (1.15, 1.71)	0.0023
Model 3 ^3^	1.86 (1.49, 2.33)	1.24 (1.01, 1.51)	1.06 (0.87, 1.29)	1.00 (Ref)	1.34 (1.10, 1.64)	0.0024
CF						
*n*	2918	2918	2918	2918	2918	
range	(3.1, 20.0)	(20.0, 25.4)	(25.4, 30.4)	(30.4, 38.2)	(38.2, 221.2)	
Median	16.3	22.9	27.8	33.6	45.1	
Cases (incidence rate, ‰ person-year)	171 (7.74)	206 (6.80)	230 (7.08)	213 (6.59)	233 (8.81)	
Model 1 ^2^	1.00 (Ref)	0.70 (0.56, 0.87)	0.63 (0.48, 0.81)	0.49 (0.35, 0.70)	0.59 (0.38, 0.90)	0.0071
Model 2 ^2^	1.00 (Ref)	0.70 (0.56, 0.88)	0.69 (0.53, 0.91)	0.62 (0.44, 0.89)	0.95 (0.61, 1.48)	0.48
Model 3 ^2^	1.00 (Ref)	0.73 (0.58, 0.91)	0.71 (0.54, 0.94)	0.65 (0.46, 0.94)	1.02 (0.66, 1.59)	0.70
Model 1 ^3^	2.03 (1.44, 2.86)	1.41 (1.07, 1.87)	1.27 (1.02, 1.57)	1.00 (Ref)	1.19 (0.96, 1.47)	0.0071
Model 2 ^3^	1.61 (1.13, 2.30)	1.13 (0.84, 1.51)	1.11 (0.89, 1.39)	1.00 (Ref)	1.53 (1.23, 1.91)	0.48
Model 3 ^3^	1.53 (1.07, 2.19)	1.11 (0.82, 1.49)	1.09 (0.87, 1.36)	1.00 (Ref)	1.56 (1.26, 1.95)	0.70
CQI						
*n*	2683	1980	2661	3772	3494	
range	(4.0, 6.0)	(7.0, 7.0)	(8.0, 8.0)	(9.0, 10.0)	(11.0, 20.0)	
Median	5.0	7.0	8.0	9.0	12.0	
Cases (incidence rate, %)	182 (8.28)	144 (7.03)	182 (7.04)	291 (7.03)	254 (7.08)	
Model 1 ^2^	1.00 (Ref)	0.84 (0.67, 1.05)	0.88 (0.72, 1.08)	0.91 (0.76, 1.10)	0.85 (0.70, 1.03)	0.83
Model 2 ^2^	1.00 (Ref)	0.89 (0.71, 1.12)	0.85 (0.69, 1.05)	0.89 (0.74, 1.08)	0.79 (0.64, 0.96)	0.35
Model 3 ^2^	1.00 (Ref)	0.90 (0.71, 1.14)	0.83 (0.63, 1.11)	0.87 (0.62, 1.24)	0.77 (0.50, 1.19)	0.30

^1^ Data were presented as HR (95% CI) estimated by using Cox proportional hazard regression models. Model 1: adjusted for age and sex. Model 2: Model 1 + education level, urbanization index, region, smoking status, alcohol consumption, BMI and physical activity status. Model 3: Model 2 + total energy, cholesterol, PUFA to SFA ratio, total carbohydrate and fiber intakes for dietary GI; Model 2 + total energy, cholesterol, PUFA to SFA ratio for dietary for CF and CQI. Abbreviations: BMI, body mass index; CF, carbohydrate to fiber ratio; CI, confidence intervals; CQI, carbohydrate quality index; GI, glycemic index; HR, hazard ratio; PUFA, polyunsaturated fatty acid; Q, quintiles; SFA, saturated fatty acid; T2DM, type 2 diabetes mellitus. ^2^ Reference group was Q1. ^3^ Reference group was Q4.

**Table 3 nutrients-15-00647-t003:** Associations between dietary GI, CF and CQI values and T2DM risk in 14,590 Chinese adults of the China Health and Nutrition Survey 1997–2015, stratified by age, sex, BMI, baseline hypertension, urbanization index, education level and dietary PUFA to SFA ratio ^1^.

Variables	*n*	Cases (Incidence Rate, ‰ Person-Year)	HR (95% CI) of Quintiles of Carbohydrate Quality Indicators					*P*- *Trend*	*P*- *Interaction*
			Q1	Q2	Q3	Q4	Q5		
Dietary GI									
Age, y									
<60	11,881	787 (6.38)	1.00 (Ref)	0.70 (0.55, 0.89)	0.62 (0.48, 0.78)	0.56 (0.43, 0.72)	0.74 (0.57, 0.97)	0.0162	0.47
≥60	2709	266 (13.13)	1.00 (Ref)	0.67 (0.46, 0.97)	0.42 (0.28, 0.64)	0.53 (0.35, 0.81)	0.69 (0.44, 1.08)	0.0311	
Sex									
Male	7402	517 (6.92)	1.00 (Ref)	0.62 (0.45, 0.85)	0.60 (0.44, 0.82)	0.54 (0.39, 0.75)	0.77 (0.55, 1.09)	0.18	0.56
Female	7188	536 (7.77)	1.00 (Ref)	0.67 (0.51, 0.90)	0.48 (0.36, 0.66)	0.50 (0.37, 0.69)	0.60 (0.43, 0.84)	0.0007	
BMI, kg/m^2^									
<24.0	9313	423 (4.51)	1.00 (Ref)	0.77 (0.55, 1.07)	0.66 (0.47, 0.93)	0.50 (0.34, 0.72)	0.94 (0.65, 1.36)	0.22	0.0259
≥24.0	5277	630 (12.62)	1.00 (Ref)	0.56 (0.43, 0.74)	0.47 (0.36, 0.62)	0.48 (0.36, 0.64)	0.52 (0.38, 0.71)	0.0002	
Baseline hypertension									
No	11,495	683 (5.80)	1.00 (Ref)	0.71 (0.55, 0.93)	0.66 (0.50, 0.86)	0.53 (0.40, 0.71)	0.72 (0.53, 0.97)	0.0116	0.52
Yes	3095	370 (14.26)	1.00 (Ref)	0.69 (0.47, 1.00)	0.49 (0.34, 0.72)	0.56 (0.38, 0.84)	0.69 (0.46, 1.04)	0.0453	
Urbanization index, median									
<69.46	7293	491 (5.86)	1.00 (Ref)	0.76 (0.47, 1.24)	0.56 (0.34, 0.91)	0.43 (0.26, 0.70)	0.45 (0.27, 0.76)	0.0004	0.0068
≥69.46	7297	562 (9.40)	1.00 (Ref)	0.58 (0.38, 0.88)	0.56 (0.37, 0.86)	0.51 (0.32, 0.81)	1.16 (0.69, 1.96)	0.75	
Education level									
Primary or lower	6860	618 (8.22)	1.00 (Ref)	0.85 (0.55, 1.32)	0.54 (0.35, 0.83)	0.46 (0.29, 0.73)	0.54 (0.33, 0.88)	0.0016	0.29
Middle or above	7730	435 (6.35)	1.00 (Ref)	0.66 (0.41, 1.05)	0.70 (0.43, 1.14)	0.53 (0.32, 0.88)	1.02 (0.59, 1.76)	0.72	
Dietary PUFA: SFA, median									
<1.15	7295	479 (6.75)	1.00 (Ref)	0.66 (0.49, 0.88)	0.62 (0.46, 0.84)	0.64 (0.47, 0.89)	1.06 (0.74, 1.51)	0.74	0.0069
≥1.15	7295	574 (7.90)	1.00 (Ref)	0.73 (0.53, 0.99)	0.58 (0.42, 0.80)	0.46 (0.33, 0.63)	0.55 (0.40, 0.76)	<0.0001	
CF									
Age, y									
<60	11,881	787 (6.38)	1.00 (Ref)	0.84 (0.64, 1.09)	0.78 (0.56, 1.08)	0.74 (0.48, 1.14)	1.31 (0.77, 2.22)	0.64	0.33
≥60	2709	266 (13.13)	1.00 (Ref)	0.47 (0.31, 0.73)	0.46 (0.28, 0.77)	0.41 (0.21, 0.79)	0.42 (0.19, 0.96)	0.0247	
Sex									
Male	7402	517 (6.92)	1.00 (Ref)	0.71 (0.51, 0.99)	0.62 (0.41, 0.94)	0.63 (0.37, 1.09)	0.85 (0.43, 1.67)	0.48	0.68
Female	7188	536 (7.77)	1.00 (Ref)	0.74 (0.53, 1.01)	0.82 (0.56, 1.20)	0.68 (0.41, 1.12)	1.19 (0.65, 2.19)	0.84	
BMI, kg/m^2^									
<24.0	9313	423 (4.51)	1.00 (Ref)	0.79 (0.54, 1.15)	0.52 (0.33, 0.82)	0.58 (0.32, 1.03)	0.60 (0.29, 1.23)	0.11	0.0081
≥24.0	5277	630 (12.62)	1.00 (Ref)	0.66 (0.50, 0.88)	0.79 (0.56, 1.13)	0.66 (0.41, 1.06)	1.41 (0.80, 2.49)	0.52	
Baseline hypertension									
No	11,495	683 (5.80)	1.00 (Ref)	0.75 (0.57, 1.00)	0.68 (0.48, 0.98)	0.62 (0.38, 0.99)	0.88 (0.48, 1.61)	0.41	0.48
Yes	3095	370 (14.26)	1.00 (Ref)	0.71 (0.48, 1.07)	0.73 (0.46, 1.17)	0.60 (0.32, 1.11)	1.08 (0.52, 2.23)	0.90	
Urbanization index, median									
<69.46	7293	491 (5.86)	1.00 (Ref)	0.69 (0.48, 1.00)	0.60 (0.37, 0.97)	0.57 (0.30, 1.09)	0.97 (0.43, 2.18)	0.88	0.44
≥69.46	7297	562 (9.40)	1.00 (Ref)	0.69 (0.51, 0.94)	0.79 (0.55, 1.13)	0.70 (0.44, 1.12)	0.86 (0.49, 1.52)	0.51	
Education level									
Primary or lower	6860	618 (8.22)	1.00 (Ref)	0.63 (0.46, 0.86)	0.60 (0.41, 0.89)	0.52 (0.31, 0.86)	0.72 (0.38, 1.35)	0.27	0.32
Middle or above	7730	435 (6.35)	1.00 (Ref)	0.81 (0.58, 1.15)	0.88 (0.58, 1.33)	0.82 (0.48, 1.41)	1.35 (0.70, 2.60)	0.63	
Dietary PUFA: SFA, median									
<1.15	7295	479 (6.75)	1.00 (Ref)	0.84 (0.59, 1.20)	0.80 (0.52, 1.23)	0.69 (0.40, 1.18)	0.93 (0.48, 1.81)	0.64	0.40
≥1.15	7295	574 (7.90)	1.00 (Ref)	0.63 (0.47, 0.86)	0.65 (0.44, 0.96)	0.57 (0.33, 0.96)	1.08 (0.57, 2.06)	0.75	
CQI									
Age, y									
<60	11,881	787 (6.38)	1.00 (Ref)	0.66 (0.51, 0.86)	0.71 (0.56, 0.91)	0.69 (0.55, 0.86)	0.63 (0.50, 0.80)	0.0015	0.0324
≥60	2709	266 (13.13)	1.00 (Ref)	1.48 (0.96, 2.26)	0.97 (0.63, 1.50)	1.23 (0.83, 1.80)	1.12 (0.75, 1.66)	0.82	
Sex									
Male	7402	517 (6.92)	1.00 (Ref)	0.71 (0.50, 1.00)	0.68 (0.49, 0.94)	0.74 (0.55, 0.99)	0.64 (0.47, 0.86)	0.0214	0.12
Female	7188	536 (7.77)	1.00 (Ref)	1.14 (0.83, 1.57)	1.04 (0.76, 1.42)	0.99 (0.75, 1.32)	0.91 (0.68, 1.21)	0.29	
BMI, kg/m^2^									
<24.0	9313	423 (4.51)	1.00 (Ref)	0.99 (0.70, 1.40)	0.90 (0.64, 1.26)	1.04 (0.77, 1.42)	0.65 (0.46, 0.91)	0.0400	0.07
≥24.0	5277	630 (12.62)	1.00 (Ref)	0.87 (0.64, 1.18)	0.78 (0.58, 1.04)	0.75 (0.58, 0.98)	0.85 (0.65, 1.10)	0.20	
Baseline hypertension									
No	11,495	683 (5.80)	1.00 (Ref)	0.83 (0.62, 1.11)	0.91 (0.69, 1.19)	0.91 (0.70, 1.17)	0.74 (0.57, 0.96)	0.08	0.47
Yes	3095	370 (14.26)	1.00 (Ref)	1.06 (0.71, 1.58)	0.71 (0.48, 1.05)	0.87 (0.61, 1.23)	0.82 (0.58, 1.17)	0.22	
Urbanization index, median									
<69.46	7293	491 (5.86)	1.00 (Ref)	0.77 (0.54, 1.09)	0.71 (0.52, 0.99)	0.81 (0.61, 1.09)	0.67 (0.49, 0.90)	0.0356	0.94
≥69.46	7297	562 (9.40)	1.00 (Ref)	0.89 (0.65, 1.22)	0.84 (0.63, 1.14)	0.81 (0.61, 1.07)	0.77 (0.58, 1.03)	0.07	
Education level									
Primary or lower	6860	618 (8.22)	1.00 (Ref)	1.14 (0.84, 1.54)	1.01 (0.76, 1.36)	1.08 (0.83, 1.42)	0.94 (0.72, 1.24)	0.055	0.0069
Middle or above	7730	435 (6.35)	1.00 (Ref)	0.61 (0.42, 0.88)	0.57 (0.40, 0.79)	0.56 (0.41, 0.76)	0.46 (0.34, 0.64)	<0.0001	
Dietary PUFA: SFA, median									
<1.15	7295	479 (6.75)	1.00 (Ref)	0.98 (0.71, 1.35)	0.96 (0.71, 1.30)	0.81 (0.60, 1.08)	0.69 (0.51, 0.94)	0.0083	0.38
≥1.15	7295	574 (7.90)	1.00 (Ref)	0.87 (0.61, 1.23)	0.79 (0.57, 1.10)	0.96 (0.72, 1.28)	0.82 (0.61, 1.10)	0.43	

^1^ Data were presented as HR (95% CI) estimated by using Cox proportional hazard regression models. Adjusted confounders included age, sex, education level, urbanization index, region, smoking status, alcohol consumption, BMI, physical activity status, total energy, cholesterol and PUFA to SFA ratio in models of CF and CQI. Total carbohydrate and fiber intakes were additionally adjusted in models of dietary GI. Abbreviations: BMI, body mass index; CF, carbohydrate to fiber ratio; CI, confidence intervals; CQI, carbohydrate quality index; GI, glycemic index; HR, hazard ratio; MET, metabolic equivalent task hour; PUFA, polyunsaturated fatty acid; Q, quintiles; SFA, saturated fatty acid; T2DM, type 2 diabetes mellitus.

**Table 4 nutrients-15-00647-t004:** Threshold analyses of the associations between dietary GI and CF and T2DM risk in 14,590 Chinese adults of the China Health and Nutrition Survey 1997–2015 ^1^.

Carbohydrate Quality Indicators	Inflection Point (95% CI)	Group	HR (95% CI)	*P*	*P-log Likelihood Ratio*
Dietary GI	72.85 (71.40, 74.05)	<72.85	0.95 (0.93, 0.96)	<0.0001	<0.0001
		≥72.85	1.11 (1.07, 1.16)	<0.0001	
CF	20.55 (17.92, 21.91)	<20.55	0.94 (0.91, 0.96)	<0.0001	<0.0001
		≥20.55	1.03 (1.02, 1.04)	<0.0001	

^1^ Data were presented as HR (95% CI) estimated by using two-piecewise Cox regression models. Adjusted confounders included age, sex, education level, urbanization index, region, smoking status, alcohol consumption, BMI, physical activity status, total energy, cholesterol and PUFA to SFA ratio in models of CF. Total carbohydrate and fiber intakes were additionally adjusted in models of dietary GI. Abbreviations: BMI, body mass index; CF, carbohydrate to fiber ratio; CI, confidence intervals; GI, glycemic index; HR, hazard ratio; T2DM, type 2 diabetes mellitus.

## Data Availability

Data described in the manuscript, code book, and analytic code will be made available upon reasonable request. Data of the China Health and Nutrition Survey are available online (http://www.cpc.unc.edu/projects/china accessed on 17 July 2020).

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
