# Peer review of "Associations between Conventional and Emerging Indicators of Dietary Carbohydrate Quality and New-Onset Type 2 Diabetes Mellitus in Chinese Adults"

_nutrients, 2023, doi:10.3390/nu15030647_

Round 1
Reviewer 1 Report
Re: "Associations between conventional and emerging indicators of dietary carbohydrate quality and new-onset type 2 diabetes Mellitus in Chinese adults"
The study's aim is important. The overall quality of the manuscript is high. The Introduction section is informative and supported by a sufficient number of references. The results section is informative and well-presented.
Please consider the following minor changes that may improve the soundness of the current paper:
1. Line 92 - please provide a citation rather than past a link to the website
2. point 2.4 "Assessment of T2DM" - please consider more suitable wording.
3. Statistical analysis - this section is very extensive and may be shortened or some parts may be moved to supplement.
4. Please provide 2-3 sentences on practical implications and further research needs.
Author Response
Comments from the Reviewer 1:
Reviewer 1: The study's aim is important. The overall quality of the manuscript is high. The Introduction section is informative and supported by a sufficient number of references. The results section is informative and well-presented.
Please consider the following minor changes that may improve the soundness of the current paper:
- Line 92 - please provide a citation rather than past a link to the website
Response: Thanks for the reviewer’s advice, a citation of the survey has been added to replace the link to the website on Page 1 Line 92.
- point 2.4 "Assessment of T2DM" - please consider more suitable wording.
Response: Consistent with the reviewer’s request, we have substituted the “Assessment of T2DM” with “Ascertainment of T2DM” on Page 3 Line 157.
- Statistical analysis - this section is very extensive and may be shortened or some parts may be moved to supplement.
Response: We thank the reviewer for pointing out this issue. In response to the reviewer’s suggestion, the statistical analysis section has been shortened, and divided into 4 subsections to improve readability. Please see details on Pages 4 Lines 210-251.
- Please provide 2-3 sentences on practical implications and further research needs.
Response: Thanks for the reviewer’s advice. Some practical implications and future research need, including “both dietary GI and CF should be controlled in a moderate range for T2DM prevention in Chinese adults” and “These findings, if replicated in a larger cohort, add new information to the current literature suggesting that there should be a reassessment of relationships between indicators of dietary carbohydrate quality and T2DM risk and careful consideration of whether these indicators should be incorporated into dietary guidelines and Nutrient Facts panel labeling intended to help reduce T2DM risk in Chinese population” have been provided in the in the conclusion section on Page 19 Lines 498-508.
Reviewer 2 Report
Please rewrite:
Line 174
In case of lack of time for dietary survey, the questionnaire survey time of the same survey year should prevail.”
Line 359
To the best of our knowledge, we for the first time demonstrated the U-shaped associations between dietary GI and CF values and T2DM risk in Chinese adults, and BMI, dietary PUFA: SFA and urbanization index significantly modified the associations between dietary GI and T2DM risk.
Authors state: " If there were inconsistent records in the different waves of follow-up, the first wave record should prevail." Seems like that needs a brief explanation in Discussion, esp. in terms of how it may have affected overall data analysis/results
Author Response
Reviewer 2:
- Please rewrite:
1)Line 174
In case of lack of time for dietary survey, the questionnaire survey time of the same survey year should prevail.”
Response: Consistent with the reviewer’s request, this sentence has been rewritten as “The baseline time was defined as the time when the participants first enrolled in the study during 1997-2015, which was the date of the dietary survey in the current analysis. In case of missing values in the date of dietary survey, the household inter-view date, which was very close to the date of dietary survey, was used as the baseline time on Page 3 Lines 173-176.
2)Line 359
To the best of our knowledge, we for the first time demonstrated the U-shaped associations between dietary GI and CF values and T2DM risk in Chinese adults, and BMI, dietary PUFA: SFA and urbanization index significantly modified the associations between dietary GI and T2DM risk.
Response: In response to the reviewer’s request, this sentence has been rewritten as follows: Data of the current study demonstrated U-shaped associations between dietary GI and CF values and T2DM risk in Chinese adults, and BMI, dietary PUFA: SFA ratio and urbanization index significantly modified the associations between dietary GI and T2DM risk. Please see details on Page 17 Lines 377-380.
- Authors state: " If there were inconsistent records in the different waves of follow-up, the first wave record should prevail." Seems like that needs a brief explanation in Discussion, esp. in terms of how it may have affected overall data analysis/results
Response: Thanks for the reviewer’s suggestion. To clarify, “the different waves of follow-up” was the survey waves of the entire CHNS survey but not the follow-up of the current study, and this statement has been clarified by using “survey waves of CHNS” to substitute “follow-up” on Page 3 Line 172. In response to the reviewer’s request, we have added one more limitation about how the inconsistent records in the different waves of follow-up may have affected overall data analysis/results on Page 19 Lines 490-497. We think the records after the first diagnosis of T2DM was not informative, because it did not capture the incident risk of T2DM and it might increase the bias from changes of lifestyle or other interventions after the diagnosis, which may undermine the association between dietary carbohydrate quality indicators and the risk of T2DM.